# Whole exome sequencing identifies *FANCM* as a susceptibility gene for estrogen-receptor-negative breast cancer in Hispanic/Latina women

Jovia L. Nierenberg[1,2], Aaron W. Adamson[3], Donglei Hu[2], Scott Huntsman[2], Carmina Patrick[3], Min Li[2], Linda Steele[3], Shu Tao[4], Yuan Chun Ding[3], Barry Tong[2], Yiwey Shieh[5], Laura Fejerman[6,7], Stephen B. Gruber[8], Christopher A. Haiman[9], Esther M. John[10,11,12], Lawrence H. Kushi[13], Gabriela Torres-Mejía[14], Charité Ricker[15], Jeffrey N. Weitzel[16], Elad Ziv[2] ✉ & Susan L. Neuhausen[3]

Breast cancer (BC) is one of the most common cancers globally. Genetic testing facilitates screening and informs targeted risk-reduction and treatments. However, genes included in testing panels are from European-ancestry studies. We conducted a pooled case-control analysis in self-identified Hispanic/Latina women (4178 cases and 4344 controls), using whole exome sequencing and a targeted panel. We tested the association of loss of function (LoF) variants with overall, estrogen receptor (ER)-positive, and ER-negative BC risk. Using logistic regression, we found a strong association of LoF variants in *FANCM* with ER-negative BC ($p = 4.1 \times 10^{-7}$), odds ratio [confidence interval]: 6.7 [2.9–15.6]). Among known susceptibility genes, *BRCA1*, *BRCA2*, and *PALB2* strongly associated with BC. *FANCM* was previously proposed as a possible susceptibility gene for ER-negative BC, but is not routinely tested clinically. Our results demonstrate that *FANCM* should be added to BC gene panels.

Breast cancer (BC) is the most common cancer in women[1], and is influenced by hormonal, environmental, and genetic factors[2]. Germline pathogenic variants found in *BRCA1*[3], *BRCA2*[4], *PALB2*[5], *TP53*[6,7] and others genes[8] are associated with high risk of BC and underlie familial cancer syndromes. Several intermediate-penetrance genes including *CHEK2*[9] and *ATM*[10] also have been identified. Genome-wide association studies (GWAS) have identified many common variants that contribute to BC risk[11]. All the pathogenic variants in susceptibility genes and the

[1]Department of Epidemiology and Biostatistics, University of California, San Francisco, San Francisco, CA, USA. [2]Department of Medicine, University of California, San Francisco, San Francisco, CA, USA. [3]Department of Population Sciences, Beckman Research Institute of City of Hope, Duarte, CA, USA. [4]Integrative Genomics Shared Resource, Beckman Research Institute of City of Hope, Duarte, CA, USA. [5]Department of Population Health Sciences, Weill Cornell Medicine, New York, NY, USA. [6]Department of Public Health Service, University of California, Davis, Davis, CA, USA. [7]UC Davis Comprehensive Cancer Center, University of California, Davis, Davis, CA, USA. [8]Department of Medical Oncology and Center for Precision Medicine, City of Hope National Medical Center, Duarte, CA, USA. [9]Department of Preventive Medicine, Norris Comprehensive Cancer Center, Keck School of Medicine, University of Southern California, Los Angeles, CA, USA. [10]Department of Epidemiology & Population Health, Stanford University School of Medicine, Stanford, CA, USA. [11]Department of Medicine, Stanford University School of Medicine, Stanford, CA, USA. [12]Stanford Cancer Institute, Stanford University School of Medicine, Stanford, CA, USA. [13]Division of Research, Kaiser Permanente Northern California, Oakland, CA, USA. [14]Instituto Nacional de Salud Pública, Cuernavaca, Mexico. [15]Department of Medicine, Keck School of Medicine, University of Southern California, Los Angeles, CA, USA. [16]Division of Precision Prevention, The University of Kansas Comprehensive Cancer Center, Kansas City, KS, USA. ✉e-mail: Elad.ziv@ucsf.edu

## Table 1 | Characteristics of study participants included in the analysis

| | Cases n = 4264 | Controls n = 4350 |
|---|---|---|
| Discovery samples, Whole-exome sequencing (Clinical Research Exome, Agilent) | n = 1043 | n = 1188 |
| CCGCRN | 885 (85%) | N/A |
| COH | N/A | 313 (26%) |
| MEC | N/A | 875 (74%) |
| UCSF | 52 (5%) | N/A |
| USC | 106 (10%) | N/A |
| Replication, targeted sequencing | n = 3221 | n = 3162 |
| CAMA | 1123 (35%) | 1122 (36%) |
| CPMCRI Cohort | 12 (0.4%) | 847 (27%) |
| PATHWAYS | 412 (13%) | N/A |
| MEC | 816 (25%) | 865 (27%) |
| NC-BCFR | 696 (22%) | 54 (2%) |
| SFBCS | 162 (5%) | 274 (9%) |
| Unselected for Hereditary Risk[a] (%) | 2993 (70%) | 3162 (73%) |
| Age (mean (SD)) | 52.1 (13) | 55.9 (12) |
| African ancestry (mean (SD)) | 0.05 (0.07) | 0.05 (0.05) |
| Indigenous American ancestry (mean (SD)) | 0.40 (0.23) | 0.43 (0.22) |
| European ancestry (mean (SD)) | 0.54 (0.22) | 0.52 (0.22) |
| Family history of breast cancer in a first degree relative (%) | | |
| Yes | 891 (21%) | 324 (7%) |
| No | 3034 (71%) | 3162 (73%) |
| Missing | 339 (8%) | 864 (20%) |
| Estrogen receptor status[b] (%) | | |
| Positive | 2094 (49%) | N/A |
| Negative | 713 (17%) | N/A |
| Missing | 1457 (34%) | N/A |
| Progesterone receptor status** (%) | | |
| Positive | 1594 (37%) | N/A |
| Negative | 940 (22%) | N/A |
| Missing | 1730 (41%) | N/A |
| Human epidermal growth factor receptor 2 status (%) | | |
| Positive | 349 (8%) | N/A |
| Negative | 1285 (30%) | N/A |
| Missing | 2630 (62%) | N/A |

*CAMA* Cancer de Mama, *CCGCRN* Clinical Cancer Genomics Community Research Network, *COH* City of Hope, *CPMCRI-Cohort* California Pacific Medical Center - Breast Health Center, *MEC* Multi-Ethnic Cohort, *PR* progesterone receptor, *UCSF* University of California San Francisco, *USC* University of Southern California.

[a] Selection was not based on hereditary risk in Kaiser, MEC, CPMC Cohort, the San Francisco Bay Area Cancer Study, CAMA, and for some participants in the Northern California Breast Cancer Family Registry.

[b]Estrogen receptor and progesterone receptor status was missing in >80% of CAMA cases and in approximately 20% of other cases.

common risk variants identified to date explain approximately half of the heritability of the disease[11].

Genetic testing for pathogenic variants in BC susceptibility genes is currently used to identify women at high risk of developing BC, who may benefit from increased screening and risk-reducing interventions, for cascade testing in families to identify other individuals at increased risk, and to inform the use of targeted treatments in those who develop cancer[12–15]. Two large, recent studies, in predominantly European ancestry participants, confirmed the association of known BC susceptibility genes with increased BC risk in population-based cohorts, reiterating that many of these genes are important to include on clinical genetic testing panels[16,17].

## Table 2 | Gene-based P-values from joint analysis of LoF variants with exome-wide significance, overall breast cancer risk in estrogen receptor-positive and estrogen receptor-negative breast cancers

| Gene | Chr | Overall | ER + | ER− |
|---|---|---|---|---|
| *BRCA1*[a] | 17 | **$2.3 \times 10^{-10}$** | 0.03 | **$4.4 \times 10^{-16}$** |
| *BRCA2*[a] | 13 | **$8.4 \times 10^{-10}$** | $3.3 \times 10^{-4}$ | **$8.0 \times 10^{-15}$** |
| *FANCM* | 14 | $9.8 \times 10^{-3}$ | 0.11 | **$4.1 \times 10^{-7}$** |
| *PALB2* | 16 | **$1.8 \times 10^{-8}$** | $1.3 \times 10^{-5}$ | $5.9 \times 10^{-5}$ |

*P*-values are from gene-based SKAT-O analyses. Bold *P*-values indicate Bonferroni corrected statistical significance, adjusting for multiple comparisons in a two-sided test at an alpha threshold of $0.05/20,000 = 2.5 \times 10^{-6}$. Overall breast cancer analyses included N = 8614 (4264 cases and 4350 controls). Estrogen receptor-positive analyses included N = 6444 independent samples (2094 cases and 4350 controls). Estrogen receptor-analyses included H = 5063 independent samples (713 cases and 4350 controls).
*Chr* chromosome; *ER* estrogen receptor; *LoF* loss of function.
[a]Discovery participants were selected for being *BRCA1/2* negative (see section "Methods"), replication results are presented for *BRCA1/2*.

Most of the knowledge of genetic susceptibility to BC is based on studies in European-ancestry populations, leaving gaps in the understanding of genetic effects in other populations. Among Hispanic/Latina (H/L) women in the United States (US), BC is the most common cancer and leading cause of cancer-related death[18]. Latin-American populations are genetically diverse, and many H/L individuals have admixed ancestry, comprised primarily of European, Indigenous American, and African components[19,20]. GWAS of BC in H/L women identified a protective variant near *ESR1*, which is most common in women with high Indigenous American ancestry[21]. In addition, unique founder variants have been identified in *BRCA1*[22,23], *PALB2*[24], and *CHEK2*[24]. To better understand the impact of rare variants in coding sequence of genes on BC risk among self-identified H/L women, we performed a whole exome sequencing (WES) and targeted replication approach in over 8500 H/L women from California and Mexico.

Here, we show an exome-wide significant association between loss-of-function (LoF) variants in *FANCM* and estrogen-receptor (ER) negative BC and a nominal association with overall BC risk. We also show significant associations between known susceptibility genes, *BRCA1*, *BRCA2*, and *PALB2* and overall BC risk, as well as nominally significant associations with *CHEK2*, *RAD51C*, and *TP53*.

## Results

The mean age at BC diagnosis was 42.6 years (standard deviation [SD]: 8.5) and the mean age of controls at enrollment was 60.3 years (SD: 10.9) in the discovery set ($P = 2.4 \times 10^{-200}$) (Table 1). In the replication set, the mean age at BC diagnosis was 55.3 years (SD: 12.0) and the mean age of controls at enrollment was 54.9 years (SD: 11.4, $P = 0.27$). European ancestry (EA) proportion was higher among cases (t = −4.53, degrees of freedom [df]= 8594.6, $P = 6.0 \times 10^{-6}$) and Indigenous American ancestry (IA) was higher among controls (t = 5.05, df = 8599.5, $P = 4.4 \times 10^{-7}$, Supplementary Fig. 1). Cases were more likely to report history of a first-degree relative with BC than controls ($\chi^2 = 99.57$, 1 df, $P = 2.5 \times 10^{-108}$). Most of the cases were ER-positive and progesterone receptor (PR)-positive and 21.3% of those tested were human epidermal growth factor receptor 2 (HER2)-positive (Table 1).

In the analysis for overall BC risk focusing on LoF variants, we found significant associations with *BRCA1*, *BRCA2* and *PALB2*, with ORs (95% CI) of 24.9 (6.1–102.5), 7.0 (3.5–14.0), and 6.5 (3.2–13.1), respectively. For ER-negative BC, LoF variants in *BRCA1*, *BRCA2* and *FANCM*, were significantly associated with ORs of 40.7 (8.9-186.5), 10.5 (4.5-24.7), and 6.7 (2.9-15.6), respectively (Table 2 and Fig. 1). Single LoF variants in each of the significant genes are shown in Supplementary Data 3. No exome-wide significant associations were found with ER-positive BC. Three other known BC genes had nominally significant associations with BC: *CHEK2* with overall and ER-positive BC, *RAD51D*

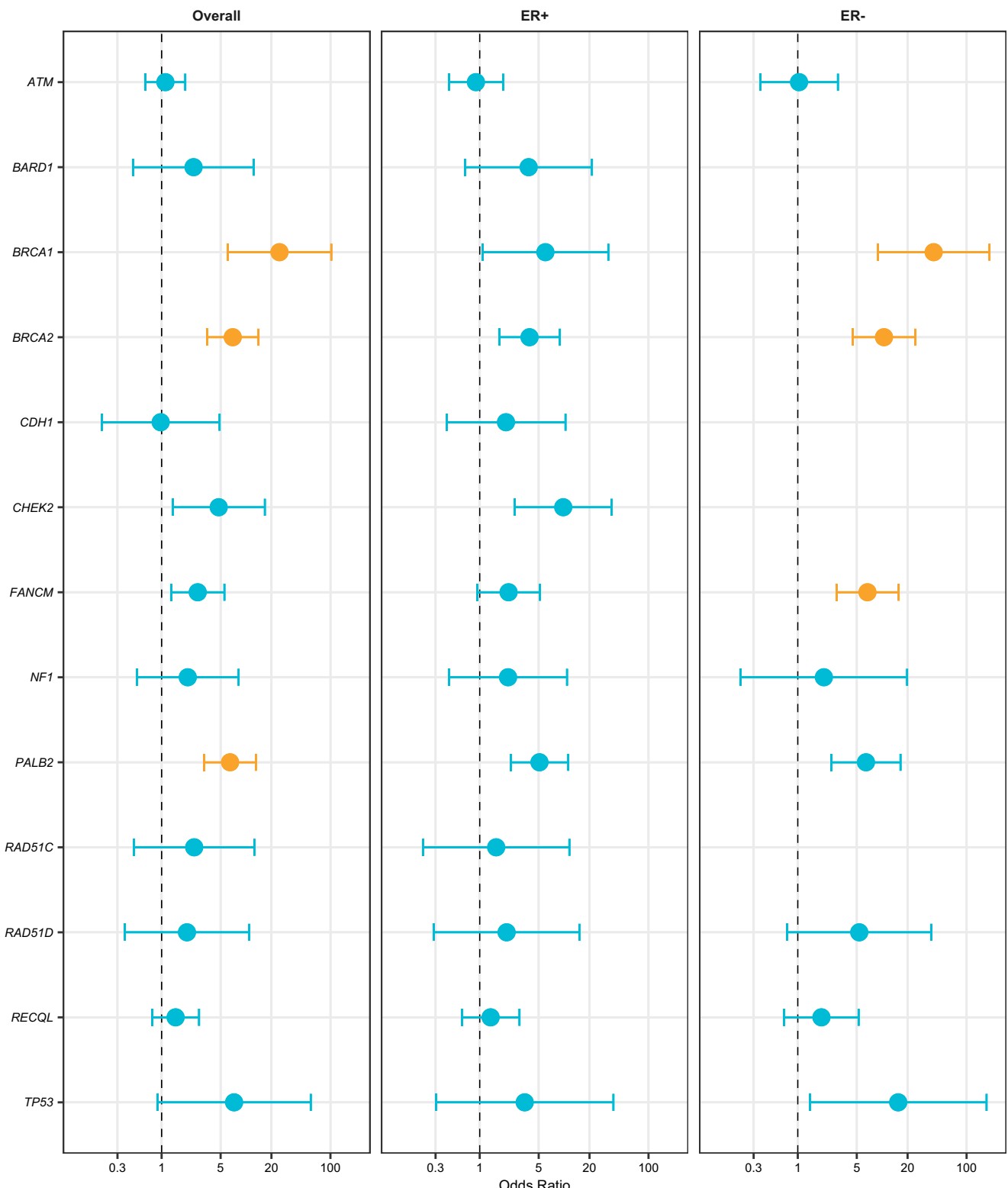

**Fig. 1 | Gene-based odds ratios from joint analysis for overall, estrogen receptor-positive, and estrogen receptor-negative disease.** Odds ratios (circles) and 95% confidence intervals (bars) are presented for overall breast cancer, (n = 8614 independent samples, including 4264 cases and 4350 controls), estrogen receptor-positive (n = 6444 independent samples, including 2094 cases and 4350 controls), and estrogen receptor-negative disease (n = 5063 independent samples including 713 cases and 4350 controls). The dot represents the point estimate and the bars represent the upper and lower bound of the 95% confidence intervals. The X-axis describes the odds ratio on a log scale, the Y-axis represents the individual genes. Statistical significance was determined using SKAT-O P-values. The exome-wide significance threshold selected was 0.05/20,000 = 2.5 × 10⁻⁶, based on 20,000 genes in the genome. Orange circles and bars represent genes with exome-wide significance. Results for a gene were included if the gene had >5 variants or at least one alternate allele in both cases and controls. Genes are listed in alphabetical order. Source data are provided as a Source Data file. ER estrogen receptor.

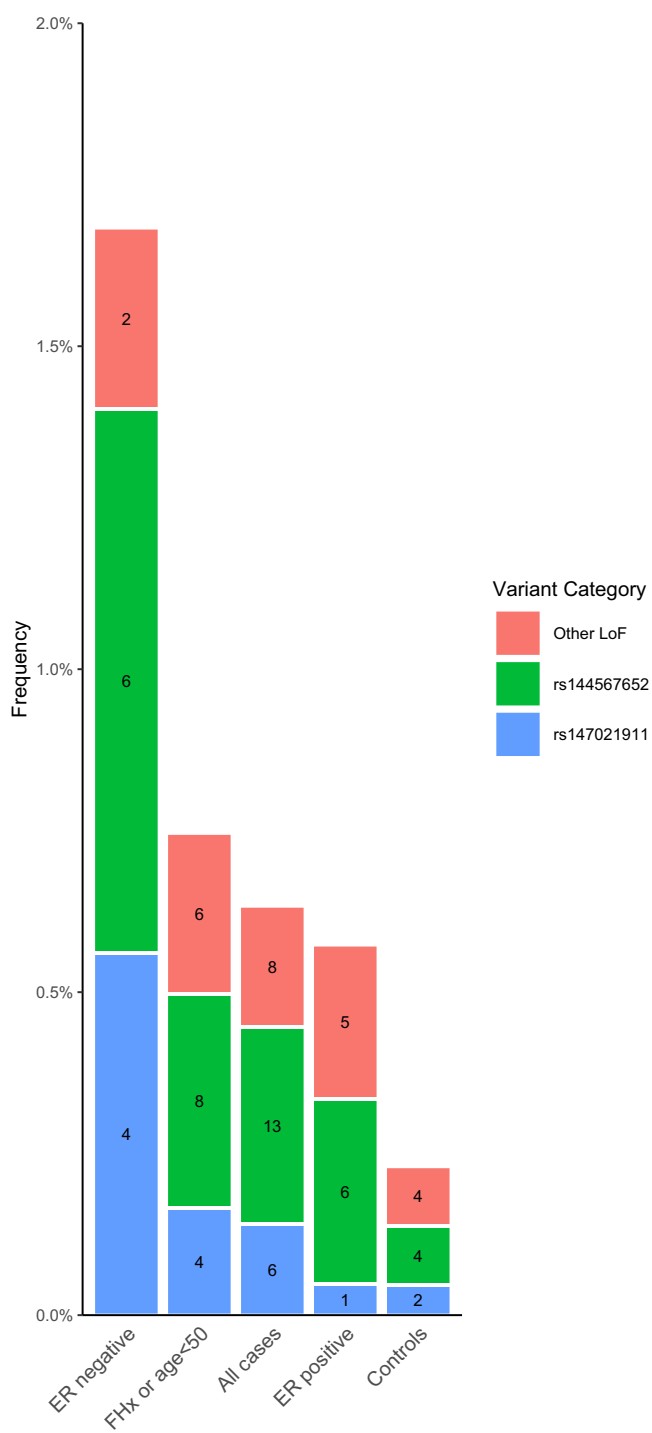

**Fig. 2 | *FANCM* loss of function (LoF) variant frequencies by participant group.** Loss of function (LoF) variants included frameshift, stopgain, and predicted splice variants. The *X*-axis shows groups of participants (n = 37 biologically independent samples with FANCM LoF variants). The *Y*-axis shows the frequency of carriers (all carriers are heterozygous). The variant, rs144567652, is also known as chr14:45667921C>A/C>T, c.5791C>T, and p.Arg1931* (overall OR = 3.2, 95% CI: 1.0–9.8 and ER-negative OR = 8.1, 95% CI: 2.3–29.0). The variant, rs147021911, is also known as chr14:45658326C>T, c.5101C>T, and p.Gln1701* (overall OR = 3.0, 95% CI: 0.6–14.7 and ER-negative OR = 11.3, 95% CI: 2.1–62.2). Other loss of function (LoF) variants found in *FANCM* include chr14:45605743G>A, rs140760056 (chr14:45633697C>T), rs368728266 (chr14:45636336C>T), rs778176467 (chr14:45642357C>T), chr14:45644584A>AT, rs1566762924 (chr14:45644795T>C), chr14:45644911, chr14:45644926TAG>T, rs755947203 (chr14:45653091C>A,T), rs930692973 (chr14:45658155C>G,T), and rs1379375089 (chr14:45665681C>T). Source data (participant counts) are provided in the figure. Source data are provided as a Source Data file. ER estrogen receptor, FHx family history of breast cancer in first-degree relatives, LoF loss of function.

(chr14:45658326C>T) and rs144567652 (chr14:45667921C>A/C>T) (Fig. 2). Among ER-negative cases, 1.7% (12/713) had a LoF variant in *FANCM*, compared to 0.6% of cases with ER-positive BC and 0.2% of controls (Fig. 2). We also analyzed the association with ER-negative, progesterone receptor (PR)-negative, Her2-negative breast cancer, frequently referred to as triple negative breast cancer (TNBC). Among the 713 women with ER negative breast cancer, 318 (45%) had missing PR or HER2 status, so TNBC status could not be determined. Among those with complete data on PR and HER2, 266 (67%) had TNBC and 129 (33%) did not. Even with this small sample size, *FANCM* was associated with TNBC (OR = 4.5, SKAT-O *p* = 0.00013).

Since ascertainment can affect the OR estimates and our discovery dataset and replication datasets were selected based on different criteria, we analyzed the top genes among replication-set participants who were from studies of cases unselected for hereditary risk (Supplementary Fig. 2). The effect sizes for *CHEK2* and *PALB2* were similar in all studies and in unselected studies. The OR for *CHEK2* and ER-positive BC in unselected studies was 10.7, which was similar to the OR of 10.8 found in all studies. We found that the association between *FANCM* and ER-negative BC was similar in studies of unselected cases (OR = 7.6) and in all studies (OR = 6.7). We also tested for interactions by age and by genetic ancestry. As expected, both rs147021911 (chr14:45658326C>T) and rs144567652 (chr14:45667921C>A/C>T) were more common among women with higher European ancestry. We found a trend towards a greater effect size among women with high Indigenous American ancestry, but this did not reach statistical significance (*p* = 0.06). We found no significant interaction between *FANCM* variants and age or study site (Supplementary Table 4).

Since many of the known genes for BC are involved in repair of double-stranded breaks, we reviewed all suggestive associations in the combined discovery and validation analyses and identified genes that are members of this pathway using a previously curated list[25]. In analyses that included both LoF variants and missense variants with high likelihood of being pathogenic, we found suggestive evidence for *ATR* and *FANCG* in analyses that included LoF and predicted deleterious missense variants (Supplementary Tables 5 and 6). *ATR* was more strongly associated with ER-positive disease, while *FANCG* was more strongly associated with ER-negative BC.

## Discussion

In this case-control study of BC in over 8,500 H/L women, we found strong evidence of association between ER-negative BC and LoF variants in *FANCM* largely driven by two *FANCM* stop-gain variants. In addition, we saw strong associations with the known BC genes *BRCA1*, *BRCA2*, and *PALB2*. *FANCM* previously had been proposed as a susceptibility gene for development of ER-negative BC although without exome-wide significance[26–29]. The two most common *FANCM* LoF

with ER-negative BC, and *TP53* with ER-negative BC (Supplementary Table 1 and Fig. 1). We found no significant associations of *ATM*, *BARD1*, *CDH1*, *RAD51C*, or *RECQL* and BC risk. We ran a sensitivity analysis for *ATM* using truncating variants only, due to the unexpectedly null results and large number of splice variants, and also found no association with BC (OR = 1.2 [0.8–1.9]). All genes with suggestive (*P* < 0.01) associations for overall, ER-positive and ER-negative BC are shown in Supplementary Tables 2 and 3. We also performed separate analyses that combined LoF variants with predicted deleterious missense variants (Supplementary Tables 5 and 6). Associations with *BRCA1*, *BRCA2* and *PALB2* remained significant, and the association with *FANCM* and ER-negative breast cancer was mildly attenuated (*p* = 2.9 × 10⁻⁶).

The association between *FANCM* LoF variants and ER-negative BC was largely driven by two stop-gain variants, rs147021911

variants we observed are more common in European populations than other populations. They were first identified in European ancestry *BRCA1/2*-negative familial BC studies[26,27], and have been associated with ER-negative or triple negative BC[28–30]. Additionally, rs147021911 (chr14:45658326C>T, c.5101C>T, p.Gln1701*) was associated with poor BC survival in a Finnish population[31], and other *FANCM* LoF mutations were found in small BC case series[29,32–35], including a report of bi-allelic *FANCM* variants in early onset or bilateral BCs[36].

Our study identified exome-wide significant association between *FANCM* and ER-negative BC in any population, but our results should be interpreted with caution. Since our sample size is relatively modest, the OR may be inflated by the winner's curse[37–39]. However, even if the OR is inflated, the higher impact on ER-negative disease is consistent with prior studies and suggests that testing for *FANCM* LoF variants can help identify women at increased risk for developing ER-negative BC. Another possibility for the high OR that we observed may be an, as yet, unknown modifying factor that affects the risk of *FANCM* and is more common among Latinas. Prior studies have identified heterogeneity among carriers of pathogenic variants in intermediate penetrance susceptibility genes including pathogenic variants in *CHEK2* which are associated with higher risk among women with more affected relatives[40]. Our study did not identify family history as a significant modifier of risk. We did note a trend towards a higher OR among women with higher Indigenous American ancestry, but this was not statistically significant. Our sample size was very limited to detect interactions. Future, larger studies of *FANCM* will be helpful to improve the precision of risk estimates associated with LoF variants in this gene and in understanding which factors may underlie any heterogeneity among carriers.

Similar to most other BC susceptibility genes, *FANCM* is involved in double-stranded DNA repair. FANCM localizes to stalled replication forks and initiates the response to double-stranded breaks by homologous recombination repair[41]. *FANCM* is a member of the Fanconi Anemia (FA) complex and is the most conserved gene within the FA pathway[42]. which is essential for handling DNA interstrand cross linking damage in DNA repair through the homologous recombination repair pathway[43]. A stronger association between *FANCM* and ER-negative disease than ER-positive disease is similar to results for *RAD51C* and *RAD51D*[16], which are also key components of the FA pathway.

Given our results and the previous associations, we believe the data support consideration of inclusion of *FANCM* on clinical testing panels. Most current genetic susceptibility breast cancer testing panels do not include *FANCM*[44] and there are no guidelines from the National Comprehensive Cancer Network on managing women who are carriers[45]. The absolute risk of breast cancer among *FANCM* LoF carriers is likely much lower than other intermediate breast cancer susceptibility genes, since ER-negative and triple negative BC constitute the minority of overall breast cancers and that is the main subtype with an increased risk. However, since the average age of diagnosis for ER-negative BC and TNBC are much lower[46], women with *FANCM* LoF variants may benefit from earlier BC screening. Moreover, women with BC and *FANCM* LoF variants may benefit from poly adenosine diphosphate-ribose polymerase (PARP) inhibitors[47], therefore, testing for *FANCM* LoF variants may identify additional therapeutic options for women with ER-negative BC.

We found that carriers of LoF variants in *CHEK2* had a particularly high risk of ER-positive disease, which is consistent with prior studies[16,17]. However, the effect sizes in our study were substantially higher than those in previous reports[16]. In analyses that excluded cases selected for familial BC, the CIs were wide but the lower bound of the CI (2.1) included the OR reported by Dorling and colleagues[16], indicating that our larger OR may be due to chance. The higher OR we observed also may be due to the younger ages of the H/L population and/or to heterogeneity of genetic or hormonal and environmental

risk factors across populations. H/L women tend to have more protective non-genetic risk factors such as younger age at first pregnancy, higher parity[48], lower postmenopausal hormone use, and lower alcohol consumption[49], when compared to US non-Hispanic White populations which may lead to stronger associations with genetic risk factors[50]. In contrast to the strong association we observed with *CHEK2*, we found no association with LoF variants in *ATM*. The upper bound of the CI for ER-positive disease excludes the reported OR in the BC Association Consortium study[16] but not the OR reported in the CARRIERS study[17]. We had previously noted no significant association with *ATM* LoF variants in Latinas[24] in a dataset that partially overlaps our current report. These results may be due to genetic or hormonal and environmental factors that attenuate the associations with *ATM* LoF variants in this population. However, until larger studies are done on *ATM* in H/L populations, it may be prudent to continue to consider these women at moderately increased risk of BC like non-H/L White women.

Our study identified strong associations with BC risk genes, including an exome-wide significant association between *FANCM* and ER-negative disease, demonstrating the importance of conducting genetic studies in understudied populations. Most large, complex trait genetic studies, including those on BC, were conducted in European-ancestry participants[51]. The lack of genetic studies in non-European populations exacerbates health disparities as genomics is increasingly used in clinical practice.

We found higher European ancestry and lower Indigenous American ancestry among US Latina and Mexican breast cancer cases compared to controls. We had previously evaluated the risk factors underlying this association using data that substantially overlap with this study[52,53]. Part of this association was explained by known reproductive and lifestyle risk factors for breast cancer. However, the association with ancestry remained significant after adjusting for these covariates[52,53]. We used admixture mapping, genome wide association, and fine mapping to identify a 6q25 variant which is associated with substantial protection from breast cancer, originates in Indigenous American populations, and explains part of the association with ancestry among Latinas[21,54].

Our study has several strengths. We focused on H/L women, a population that is understudied for breast cancer risk. We included participants from both hereditary-risk studies and unselected cases and controls, and compared results across these two study types, making our findings applicable to both groups. Our study also has several limitations. The WES for discovery was performed in only 1043 cases and 1188 controls and was likely underpowered to find intermediate-penetrance genes. To compensate for this, we sequenced 857 genes for replication in 3221 cases and 3162 controls. This sample size is still much lower than those used in discovery and validation of breast cancer susceptibility genes in European ancestry populations[55]. Larger studies in H/L populations are needed to provide better point estimates, to identify new candidate genes for BC, and to detect and understand factors contributing to heterogeneity of effect sizes compared to studies in US non-Hispanic White and European populations.

Our discovery phase included sequencing data previously collected from a subset of controls in the MEC using a slightly different WES target capture kit. This difference could lead to spurious associations due to technical differences between the different capture kits or to demographic differences in the discovery set cases and controls However, we accounted for these potential differences by adjusting for ancestry and excluding variants that were significantly different in an analysis of our two discovery control populations.

In conclusion, our study demonstrates an exome-wide significant association between LoF variants in *FANCM* and ER-negative BC in H/L women from multiple studies. We also found exome-wide significant associations for *BRCA1*, *BRCA2*, and *PALB2*. Our findings suggest that

*FANCM* should be added to genetic testing panels for BC, which is especially important for H/L women. Additionally, our findings demonstrate the importance of conducting genetic studies in admixed populations.

## Methods

### Study samples

The research complies with all relevant ethical regulations.All participants were consented using written signed forms and enrolled through center-specific institutional review board-approved protocols. The institutional review boards included the City of Hope Beckman Research Center, the University of California, San Francisco, Stanford University, Kaiser Permanente Northern California, Division of Research, the University of Southern California and the National Institute of Public Health, Cuernavaca, Mexico. We performed discovery and replication in a pooled case-control study of invasive BC among self-identified H/L women in the U.S. and women recruited in Mexico. Cases had been diagnosed with at least one invasive BC and we used age at first BC diagnosis for women diagnosed with two or more metachronous BCs[23]. Discovery cases were selected for having previously tested negative for *BRCA1/2*, and were diagnosed at <51 years of age and/or had bilateral (synchronous or metachronous) BC, breast and ovarian cancers, or were diagnosed between 51 and 70 years with a family history of BC in ≥2 first-degree or second-degree relatives diagnosed at age <70 years[24]. H/L participants in the replication set were from six studies (Supplemental Methods). We included 8,614 participants in our study and the mean (standard deviation) ages of cases and controls were 52.1 (13) and 55.9 (12) years old.

### Ethics & inclusion statement

The research study has included local researchers throughout the research process including study design, data acquisition, data ownership and authorship. The local researchers deemed this research relevant. Roles and responsibilities were agreed upon amongst the collaborators before the research, and capacity building for local researchers was discussed. Local institutional review boards evaluated the research and approved. Participants were consented for research testing of genetic/inherited risk of breast cancer. Local research was considered as part of citations.

### Sequencing and genotyping

Whole exome sequencing (WES) from DNA of discovery participants was performed using the SureSelect Clinical Research Exome (Agilent, Santa Clara, CA) kit[24]. For participants in the replication set, 857 genes were selected for targeted sequencing based on the discovery results and biological plausibility. A custom SureSelect XT kit (Agilent, Santa Clara, CA) was used to capture the exons of 857 genes (Supplementary Data 1) and also included 189 known BC single nucleotide polymorphisms (SNPs) and 100 ancestry informative SNP markers. For all sequencing, we used KAPA Hyper Preparation Kits (Kapa Biosystems, Inc., Wilmington, MA) to generate libraries from 500 ng DNA. One hundred base-pair paired end sequencing on the HiSEQ 2500 Genetic Analyzer (Illumina Inc., San Diego, CA) was performed in the COH Integrative Genomics Core (IGC) to an average fold coverage of 65× for the WES samples and ~75× for the targeted-sequencing samples (Supplementary Data 2). Paired-end reads were aligned to human reference genome (hg37) using the Burrows-Wheeler Alignment Tool (BWA, version 0.7.5a-r405) under default settings, and the aligned binary format sequence (BAM) files were sorted and indexed using SAMtools[56,57]. The same FASTA reference file was used to align the MEC control samples. Duplicate reads were removed from the sorted and indexed BAMs using Picard MarkDuplicates (version 1.67, http://broadinstitute.github.io/picard/).

Variant calling from BAM files from COH IGC and the Broad Sequencing Center were performed together using GATK HaplotypeCaller (https://software.broadinstitute.org/gatk) after local realignment of reads around insertions and deletions (indels) and base quality score recalibration by The Genome Analysis Toolkit (GATK, v3.6-0-g89b7209). Variants were annotated using ANNOVAR[58]. We excluded participants with <20-fold average coverage. Variants with call quality <20, read depth <10, less frequent allele depth of <4, or allele fraction ratio <30% were filtered out for low quality. DNA from eight MEC participants were sequenced at both COH IGC and the Broad Sequencing Center with >99.8% concordance for variant calling.

From the discovery dataset (whole exome sequencing), we excluded 14 samples due to insufficient coverage (<20X), 4 duplicate samples, and 17 samples who were first- or second-degree relatives. We used PLINK 1.9 (http://www.cog-genomics.org/plink/1.9/) 28 to exclude first- or second-degree relatives or duplicate samples within the discovery and replication samples. We excluded an additional eight participants from the discovery analysis who were identified to have previously undetected *BRCA1/2* pathogenic variants. We also excluded 73 participants from the replication set who were duplicates from the discovery set. After all exclusions, there were 4260 cases and 4350 controls (Table 1).

Due to possible differences among samples sequenced at the Broad and COH in our discovery sample, we conducted a control-control analysis and excluded variants that were different between the Broad controls (from the Multi-ethnic cohort) and COH control populations at $p < 0.05$. We also conducted a sensitivity analysis within the CCGCRN/COH only, and excluded variants where the sensitivity analysis point estimate was outside of the 95% confidence interval of the main analysis.

### Ancestry estimation

The Clinical Research Exome included a custom panel of 180 ancestry-informative SNPs selected to be informative for European, Indigenous-American, and African populations[59]. In addition, we selected 7691 variants common to our WES data and a data set of Axiom arrays, including African (N = 90), European (N = 90), and Indigenous American (N = 71) populations. We used a subset of 4544 unlinked markers by linkage disequilibrium pruning in PLINK[60]. We estimated genetic ancestry using ADMIXTURE 1.3[61] and performed analyses with unsupervised (including data from ancestral populations, but not specifying the identity of ancestral populations) runs. The same approach was used for the MEC control participants where we selected independent variants (n = 12,758) that overlapped between the Axiom arrays and the MEC dataset and in the replication set where we selected 1,195 SNPs that overlapped across all replication and reference datasets.

### Statistical analysis

Gene-based aggregate rare variant analyses were based on loss of function (LoF) variants, including frameshift, stop gain, and predicted splice variants. Variants with minor allele frequency >0.0025 in our control study population and benign clinical significance in CLINVAR[62] were excluded from the gene-based analyses. Statistical significance for each gene was determined using SKAT-O[63,64]. We used a *p*-value of $2.5 \times 10^{-6}$ for the threshold of genome-wide significance which adjusts for ~20,000 coding sequence genes as has been done previously. Odds ratios (OR) and 95% confidence intervals (CI) for BC associated with any LoF variant for each gene were calculated using logistic regression models in which women with at least one LoF variant in a gene were encoded as "1" and all other women were encoded as "0". These models included ancestry (European and Indigenous American) as covariates and constituted our burden tests. We only included genes that had >5 variants or at least one alternate allele in both cases and controls. We used the replication sample for all *BRCA1/2* analyses, as *BRCA1/2* status was an exclusion criterion for the discovery sample. We looked for the known Mexican founder *BRCA1* exon 9–12 deletion[23], and verified its presence using the Integrative Genomics Viewer (IGV),

and included this variant in analyses, but did not perform additional large deletion analyses. In addition, to LoF variants, we also performed separate analyses that included missense variants that were predicted to be deleterious based on VEST[25] with VEST score >0.8 which were analyzed in combination with LoF variants.

Single variant analyses and aggregate rare variant analyses were conducted in discovery and replication separately and a joint discovery and replication analysis. BC subtype-specific analyses were conducted, where cases were restricted to ER-positive and ER-negative disease. Sensitivity analyses were run separately for participants from hereditary BC studies and studies that did not select cases based on BC risk factors.

To test for interactions with *FANCM* variant status, we used logistic regression models and entered presence of a *FANCM* LoF variant multiplied by either age or genetic ancestry as continuous variables or family history as a dichotomous variable.

### Reporting summary

Further information on research design is available in the Nature Portfolio Reporting Summary linked to this article.

## Data availability

Source data are provided with this manuscript. The genotype data and raw sequencing readsgenerated in this study have been deposited in the dbGAP database and are available at phs003144.v1.p1 [https://www.ncbi.nlm.nih.gov/projects/gap/cgi-bin/study.cgi?study_id=phs003144.v1.p1]. The genotype data are available under restricted access due to the informed consent under which the sample were obtained. Access can be obtained be obtained by applying with a scientific proposal to dbGAP. Source data are provided with this paper.

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

## Acknowledgements

We thank the participants of the SFBCS/NC-BCFR, Pathways, MEC, CAMA, and COH-CCGCRN studies. This work was funded by the NIH/NCI (R01CA184585, K24CA169004 to E.Z.) and the State of California Initiative to Advance Precision Medicine (OPR18111 to EZ, SN and L.F.). Research reported in this publication included work performed in the City of Hope Integrative Genomics Core supported by the NIH/NCI under grant number P30CA033572. The content and views are solely the responsibility of the authors and should not be construed to represent the views of the National Institutes of Health. S.L.N. and this research were partially funded by the Morris and Horowitz Families Professorship. The Multiethnic Cohort was supported by NIH/NCI grants (R01CA164973, R01CA054281, and CA063464 to C.H.). Sequencing of the MEC controls was conducted as part of the Slim Initiative for Genomic Medicine, a project funded by the Carlos Slim Health Institute in Mexico. Seventy-six of the controls were from the California Teachers Study; the collection and data were funded through the NIH/NCI (R01CA77398). The Northern California Breast Cancer Family Registry was supported by a grant from the National Cancer Institute (UM1 CA164920 (to E.J.). The SFBCS was funded by grants from the NIH/NCI (R01CA063446 and R01CA077305 to EJ), grant from the US Department of Defense (DAMD17-96–1-6071 to E.Z.), and a grant from the California Breast Cancer Research Program (7PB-0068 to E.J.). The Kaiser Permanente Research Program on Genes, Environment and Health was supported by Kaiser Permanente national and regional Community Benefit programs, and grants from the Ellison Medical Foundation, the Wayne and Gladys Valley Foundation, and the Robert Wood Johnson Foundation. The PATHWAYS cohort was supported by grants from the National Cancer Institute (U01 CA195565 R01CA105274 to LK) The CAMA Study was funded by Consejo Nacional de Ciencia y Tecnologıa, Mexico (SALUD-2002-C01–7462 to G.T.M.) and recruitment at the Puebla site was supported by the National Cancer Institute (R01CA120120 to E.Z.). Additional support was provided the NIH/NCI R01CA204797 (to L.F.). Funding to recruit women from for the CCGCRN includes Breast Cancer Research Foundation grant #20-172 (to J.N.W.), and American Society of Clinical Oncology Conquer Cancer® Research Professorship in Breast Cancer Disparities (to J.N.W.) and additional funding was provided by NIH/NCIH RC4 CA153828 (to J.N.W) Support was also provided from National Center for Advancing Translational Sciences under award KL2TR001870 (to YS) and the National Cancer Institute under award K08CA237829 (to Y.S.).

## Author contributions

E.Z. and S.N. designed the study. J.L.N., E.Z. and S.N. drafted the manuscript. J.L.N., D.H., and S.H., conducted statistical analyses. E.Z., S.N., J.L.N., S.T., B.T., S.B.G., C.A.H., E.M.J., L.H.K., G.T., C.R., and J.N.W. contributed samples and data. A.W.A., C.P., and M.L. prepared samples

for sequencing. L.S. performed clinical data ascertainment and cleaning. Y.C.D., Y.S. and L.F. made critical revisions to the manuscript. All authors reviewed and approved the final manuscript.

## Competing interests

J.L.N. is an employee and shareholder at BridgeBio Pharma, this work was done prior to her work at BridgeBio Pharma. J.N.W. is a consultant to Natera, MyOme and Cancer IQ, holds equity in Natera and is a co-founder of Novi Health. All other authors declare no competing interests.
