## [Peer Review File · Nature Communications]

Whole exome sequencing identifies FANCM as a susceptibility gene for estrogen-receptor-negative breast cancer in Hispanic/Latina women

Corresponding Author: Professor Elad Ziv

Version 0:

Reviewer comments:

Reviewer #1

(Remarks to the Author)

The manuscript titled: "Whole exome sequencing identifies FANCM as a susceptibility gene for estrogen-receptor-negative breast cancer in Hispanic/Latino women" performed a case-control analysis (4,178 breast cancer cases and 4,344 controls) using whole exome or targeted sequencing to identify breast cancer susceptibility genes.

Specific Comments

- The novelty of this study is finding that LoF variants in FANCM are associated with risk of ER-negative breast cancer in H/L women. Previous studies (such as 10.1038/s41523-019-0127-5 and <https://doi.org/10.1073/pnas.140790911>) have linked FANCM to breast cancer risk, also the current study shows that the majority of the risk is largely linked to two stopgain variants which were previously known (rs147021911 and rs144567652). Therefore, the main novelty from the current study is confirming the association of FANCM in H/L women.
- The authors propose that FANCM should be included on genetic testing panels for BC. Due to the previous studies suggesting FANCM as a cancer gene, is FANCM currently included on any genetic testing panels for BC?
- The average fold depth of sequencing was provided "an average fold coverage of 65x for the WES 152 samples and ~75x for the targeted-sequencing samples". The methods also mention that some samples were excluded due to sequence statistics. Inclusion of a supplementary table with sequence depth, type of sequencing (exome or panel), other sequence quality metrics for inclusion and clinical features for each participant would be very useful. Clinical features should include age of first cancer, cancer type, histopathology status, and any other features that may be useful to the reader.
- The methods include "Gene-based aggregate rare variant analyses were based on loss of function (LoF) variants, including frameshift, stop gain, and predicted splice variants". – were missense variants that were predicted LoF also included in the analysis? As some missense variants may have bioinformatic predictor scores suggesting LoF or are reported as likely pathogenic or pathogenic in Clinvar, please provide a rationale for not including them in the analysis. Missense variants "with high likelihood of being pathogenic" were included the latter analysis only of DNA repair genes. Apologies if I missed this, but please include in the methods how these were considered as likely pathogenic.
- Additionally, please provide more information on: "Variants with minor allele frequency > 0.0025" - which population data was used for this, and were any known founder variants identified in the cohorts?
- Table 1 summarizes all participants, however, some samples were excluded from analysis. The exact number of participant used in each analysis is not clear. Therefore, it would be good to show in table 1 the number of samples which were exclude from analysis, or perhaps more preferable only include samples that were used in the analysis in Table 1.
- What are the reasons for a higher European ancestry proportion in cases, and a higher Indigenous American ancestry in controls?
- Have the variants identified in this study been submitted to Clinvar? This would be clinically useful. Alternatively, predicted LoF variants in genes with an increase risk of BC could be provided in a supplementary table. Although some of this information is provided in the legend of Figure 2, more detailed information for each variant in a supplementary table would be useful. This would include genome coordinates of variant, variant change, annotated transcript, exon, clinvar information, population information, any relevant bioinformatic predictor scores eg CADD.
- Was FANCM also associated with participants with TNBC cases (as previously reported doi:10.1038/s41523-019-0127-5)? What proportion of the ER neg cases are TNBC?

- Please check reference 48, as I could not find this citation.
- Has the sequence data been deposited so that other researchers can access the data?

Reviewer #2

(Remarks to the Author)

The authors performed a case-control study to search for genetic susceptibility variants in coding regions in a large data set from different cohorts focused on characterizing the Hispanic-Latino population. This work contributes to close the gap in the knowledge of breast cancer in human populations, especially in those less studied compared to European populations such as Hispanic-Latinos.

I believe that good data harmonization was applied to limit bias between the different study cohorts using a dedicated pipeline. Although I recognize that this effort is one of the largest studies aiming to characterize susceptibility variants in Hispanic/Latina women, and most of their results confirmed well-known mutational profiles associated with breast cancer risk, the more novel data, unfortunately, are limited by statistical power, and therefore I believe that some aspects of the study could be improved.

1. The authors described the FANCM mutation as a relevant variant with a significantly increased risk of breast cancer in ER-negative cases, and therefore propose to include this gene in clinical panels. However, as the authors also commented, the LOF mutation in FANCM had been previously described as a possible susceptibility gene in the European population and, moreover, these variants are more frequently described in European populations. How to reconcile these data in the context of Hispanic Latina women?
2. Although the analyses were performed by ancestry adjustment, I believe that further characterization of these variants and ancestry components should be done to get a better idea of the relationship of these variants to the American Indian component. I would ask to include an analysis of the distribution of the described variants between ancestry groups (i.e. individuals with 0-25%, 25-50%, 50-75% and 75-100% American Indian component), by age (i.e.: >45 vs <45 years), by study/city, this analysis could provide some clues about environmental risk factors between populations, mainly between U.S. Mexicans and Mexicans living in Mexico.
3. The authors clearly acknowledge one of the main limitations of the study in relation to the sample size that could inflate the statistical analysis. It means that of the 713 ER-negative cases 1.7% of the patients had a LoF variant in FANCM, those 12 patients. I believe that this low number of patients is an important limitation to any biological or clinical conclusion, and should not only be stated in the discussion section, but should be treated as such throughout the text.
4. If the authors can provide a description of the allelic status of the FANCM variation (monoallelic vs. biallelic), this could provide new insights into loss of heterozygosity (LOH) phenotypes in this alteration.
5. Since the authors have access to hereditary risk studies, it would be interesting to see the status of the LoF variant of FANCM in these patients.
6. Would the inclusion of inactivating and pathogenic missense variants along with the nonsense mutations (frameshift, stop-gain, and splice variants) somehow improve power analysis?
7. There is no analysis of risk estimation or gene identification by age group of case/control matches. Some of these genetic factors may be enriched among younger women, and the lack of analysis of age as a contributing factor to gene identification may preclude the identification of other genes.
8. Given that exosome sequencing is available to the authors, I wonder why they have not considered evaluating copy number alterations (i.e. deletions). For example, about 10% of BRCA1 mutations are due to large deletions, this approach could also help improve the identification of some novel genes.
9. I believe that the descriptive analysis of the variant landscape could be enriched with mutually altered profiles.
10. How is the frequency of somatic variations in FANCM in ER- tumors?, there are also some public tumor data from various populations (including Hispanic latino) that could be included in this analysis to partially recapitulate what was observed at the germline level.

Version 1:

Reviewer comments:

Reviewer #1

(Remarks to the Author)

All of my questions were answered.

Reviewer #2

(Remarks to the Author)

The authors responded optimally to most of my comments, improving the conclusions of their work with respect to the heterogeneity and variation across phenotypes of the patients evaluated. While I still consider the study sample to be a small sample, which has a very important impact on the statistical and biological power of the conclusions, I consider it an important effort to remedy the lack of inclusion of Hispanic Latino populations. Therefore, I consider that the article can be publishable.

Reviewer 1

Specific Comments

- The novelty of this study is finding that LoF variants in FANCM are associated with risk of ER-negative breast cancer in H/L women. Previous studies (such as 10.1038/s41523-019-0127-5 and <https://doi.org/10.1073/pnas.140790911>) have linked FANCM to breast cancer risk, also the current study shows that the majority of the risk is largely linked to two stopgain variants which were previously known (rs147021911 and rs144567652). Therefore, the main novelty from the current study is confirming the association of FANCM in H/L women.

Response: *We agree with the reviewer that the main results demonstrate an overall effect of FANCM which is driven by two previously described variants and several other variants. However, our study is the first, to our knowledge, to identify an exome-wide level of significance. Some of this may be due to winner's curse, which we acknowledge, and some may be due to unknown modifiers which we discuss in the more detailed discussion in our revised manuscript.*

- The authors propose that FANCM should be included on genetic testing panels for BC. Due to the previous studies suggesting FANCM as a cancer gene, is FANCM currently included on any genetic testing panels for BC?

Response: *We have added more details about FANCM coverage on genetic testing panels in the discussion.*

- The average fold depth of sequencing was provided "an average fold coverage of 65x for the WES 152 samples and ~75x for the targeted-sequencing samples". The methods also mention that some samples were excluded due to sequence statistics. Inclusion of a supplementary table with sequence depth, type of sequencing (exome or panel), other sequence quality metrics for inclusion and clinical features for each participant would be very useful. Clinical features should include age of first cancer, cancer type, histopathology status, and any other features that may be useful to the reader.

Response: *We have added a table with coverage for each sample. Primary data on clinical variables along with primary genotype data are being deposited in dbGAP (see below).*

- The methods include "Gene-based aggregate rare variant analyses were based on loss of function (LoF) variants, including frameshift, stop gain, and predicted splice variants". – were missense variants that were predicted LoF also included in the analysis? As some missense variants may have bioinformatic predictor scores suggesting LoF or are reported as likely pathogenic or pathogenic in Clinvar, please provide a rationale for not including them in the analysis. Missense variants "with high likelihood of being pathogenic" were included the latter analysis only of DNA repair genes. Apologies if I missed this, but please include in the methods how these were considered as likely pathogenic.

Response: *We used the same general approach of the Breast Cancer Association Consortium (PMID: 33471991) and CARRIERS (PMID: 33471974) studies. We first performed analyses with only proven LoF variants and these results are the main results presented in Figures and Tables. Similar to the BCAC and CARRIERS study, we performed analyses of variants that included LoF variants and missense variants predicted to be deleterious based on VEST (>0.8), as described in the methods. These analyses are presented as supplementary results (Supplementary Tables 4-5) as they did not yield any additional significant results. We have added additional details in the results and discussion about missense variants.*

- Additionally, please provide more information on: "Variants with minor allele frequency > 0.0025" - which population data was used for this, and were any known founder variants identified in the cohorts?

Response: *We used a threshold of 0.0025 in the control population from our sample set and these variants were only excluded if they had benign clinical significance in Clinvar. We updated the text to specify that this was in our own study population.*

- What are the reasons for a higher European ancestry proportion in cases, and a higher Indigenous American ancestry in controls?

Response: *We have previously examined higher European ancestry in several publications (PMID: 19047150; PMID: 20332279 PMID: 22228098 PMID: 25327703). We have demonstrated that this is due in part to non-genetic risk factors (PMID: 19047150; PMID: 20332279) but that there is a residual association with ancestry and this is explained, in part by a locus identified by “admixture mapping” (PMID: 22228098). Fine mapping of the admixture mapping locus identified a genetic variant associated with substantial protection from breast cancer is more common in Latinas with more Indigenous American ancestry and explains part of the higher risk in European ancestry women (PMID: 25327703). We have added this to the discussion.*

- Have the variants identified in this study been submitted to Clinvar? This would be clinically useful. Alternatively, predicted LoF variants in genes with an increased risk of BC could be provided in a supplementary table. Although some of this information is provided in the legend of Figure 2, more detailed information for each variant in a supplementary table would be useful. This would include genome coordinates of variant, variant change, annotated transcript, exon, clinvar information, population information, any relevant bioinformatic predictor scores eg CADD.

Response: *We have added Supplementary Table 1 with all of the LoF variants in each of the significant BC risk genes, and their counts in cases and controls.*

- Was FANCM also associated with participants with TNBC cases (as previously reported doi:10.1038/s41523-019-0127-5)? What proportion of the ER neg cases are TNBC?

Response: *Among the 713 women with ER-negative breast cancer, 318 (45%) have missing PR or HER2 status, so TNBC status cannot be determined. Among those with complete data on PR and HER2 status (n=395), 266 (67%) have TNBC and 129 (33%) do not. Even with this small sample size, FANCM was suggestively associated with TNBC (OR=4.5, SKAT-O p=0.00013). We have added this result to the manuscript with the caveat that due to the rate of missing data on Her2 status, our estimate of the association has some uncertainty.*

- Please check reference 48, as I could not find this citation.

Response: *Reference 48 is an abstract from the 2022 ASCO Annual Meeting. The link to the abstract is: https://ascopubs.org/doi/10.1200/JCO.2022.40.16_suppl.3124. However, since it has not been published, we have omitted it from the revised manuscript; citations of peer-reviewed, published full manuscripts are used to support the claim.*

- Has the sequence data been deposited so that other researchers can access the data?

Response: *We are currently submitting to dbGAP. We intend to make dbGAP access coincident with publication of the results.*

Reviewer 2:

1. The authors described the FANCM mutation as a relevant variant with a significantly increased risk of breast cancer in ER-negative cases, and therefore propose to include this gene in clinical panels. However, as the authors also commented, the LOF mutation in FANCM had been previously described as a possible susceptibility gene in the European population and, moreover, these variants are more frequently described in European populations. How to reconcile these data in the context of Hispanic Latina women?

Response: *We agree that the question of how to reconcile these is important. We note that: (a) nearly all studies demonstrate ER-neg BC excess which is consistent with our results; (b) heterogeneity may be present as it is for other genes which appear to have variable penetrance based on other risk factors (e.g. CHEK2 with family history); and/or (c) that the strong odds ratio that we detect may be inflated by winner’s curse. To address the question of heterogeneity, we performed additional heterogeneity analyses (see response to #2 below) and we noted a non-*

significant trend towards heterogeneity by ancestry. We expanded the discussion about sources of heterogeneity and also about the possibility of winner's curse.

2. Although the analyses were performed by ancestry adjustment, I believe that further characterization of these variants and ancestry components should be done to get a better idea of the relationship of these variants to the American Indian component. I would ask to include an analysis of the distribution of the described variants between ancestry groups (i.e. individuals with 0-25%, 25-50%, 50-75% and 75-100% American Indian component), by age (i.e.: >45 vs <45 years), by study/city, this analysis could provide some clues about environmental risk factors between populations, mainly between U.S. Mexicans and Mexicans living in Mexico.

Response: *We agree. We have added analyses to evaluate the effect of ancestry, age, family history (Supplementary Table 5). We could not test interactions with country since the vast majority of cases from Mexico did not have ER-status. For ancestry we used the categories of 0-25%, 25-50%, and >50% since there were too few women with >75% Indigenous American ancestry. We did not find any significant interactions. We did note a trend towards an interaction with ancestry ($p=0.06$). The association with the lowest Indigenous American ancestry group (<25%) was closer to the association previously described in European ancestry populations while the association for women with higher Indigenous ancestry was stronger. Therefore, it is possible that an interaction may exist with ancestry (which would in turn suggest other genetic and environmental modifiers since ancestry itself is not likely to be causal). However, our study is under-powered to detect interactions. Therefore, we have added comments about the possibility of interactions in the discussion section, but we did not speculate about the interaction with ancestry since we believe that larger studies will be needed to determine if this interaction is statistically significant.*

3. The authors clearly acknowledge one of the main limitations of the study in relation to the sample size that could inflate the statistical analysis. It means that of the 713 ER-negative cases 1.7% of the patients had a LoF variant in *FANCM*, those 12 patients. I believe that this low number of patients is an important limitation to any biological or clinical conclusion, and should not only be stated in the discussion section, but should be treated as such throughout the text.

Response: *We have added the absolute numbers in the results section and additional caveats to the discussion.*

4. If the authors can provide a description of the allelic status of the *FANCM* variation (monoallelic vs. biallelic), this could provide new insights into loss of heterozygosity (LOH) phenotypes in this alteration.

Response: *The germline variants that we evaluated are all germline heterozygous since they are rare. We did not include data on somatic mutations as this is beyond the scope of this paper which is focused on germline susceptibility.*

5. Since the authors have access to hereditary risk studies, it would be interesting to see the status of the LoF variant of *FANCM* in these patients.

Response: *We agree that this is an interesting question. We evaluated the potential for interaction with family history (which was present in both cases from hereditary cancer studies and in unselected cases). Of note, among women with a family history and ER-negative breast cancer there were 4 cases with *FANCM* LoF variant and there were none among controls with family history. However, we did not see an interaction (Supplementary Table 5). These results should be interpreted with caution based on very small sample sizes.*

6. Would the inclusion of inactivating and pathogenic missense variants along with the nonsense mutations (frameshift, stop-gain, and splice variants) somehow improve power analysis?

Response: *Pathogenic missense variants only were relevant in known BC genes and were not relevant for *FANCM*. We did conduct additional analyses based on predicted deleterious missense variants (as noted in response to comments from reviewer 1). We included these results in the revised manuscript. They did not have a substantial effect on statistical significance.*

7. There is no analysis of risk estimation or gene identification by age group of case/control matches. Some of these genetic factors may be enriched among younger women, and the lack of analysis of age as a contributing factor to gene identification may preclude the identification of other genes.

Response: *We have performed analyses by age subgroups as requested (see replies to comment 2).*

8. Given that exosome sequencing is available to the authors, I wonder why they have not considered evaluating (i.e. deletions). For example, about 10% of BRCA1 mutations are due to large deletions, this approach could also help improve the identification of some novel genes.

Response: *We included the BRCA1 large deletion known in H/L women. We have clarified this in the revised manuscript. Beyond this known founder mutation, identification of other structural variants is difficult with targeted short read (Illumina) sequencing and has not been done in other large sequencing projects (BCAC, CARRIERS), likely for this reason. We added this limitation to our discussion.*

9. I believe that the descriptive analysis of the variant landscape could be enriched with mutually altered profiles.

Response: *Variants were extremely rare and more than one variant in an individual was not present in the dataset. Therefore, we did not include variant landscape analysis.*

10. How is the frequency of somatic variations in FANCM in ER- tumors?

Response: *We agree that analyses of somatic copy number loss, somatic mutation and/or epigenetic silencing would be of interest. However, we feel that large scale analyses of the somatic landscape in FANCM carriers is beyond the scope of the current manuscript which focuses on germline analyses.*

11. There are also some public tumor data from various populations (including Hispanic latino) that could be included in this analysis to partially recapitulate what was observed at the germline level.

Response: *To our knowledge the only publicly available dataset that has H/L women with breast cancer is All of US. We evaluated whether All of US has ER-status on breast cancer cases. We found only 93 of 805 breast cancer cases among Latina women have ER-status and only 20 of these 93 are ER-negative. These numbers are insufficient to attempt replication.*